# Steal the Patch Size: Adversarially Manipulate Vision-Language Models

**Kai Hu** [1]  **Akash Bharadwaj** [1]  **Weichen Yu** [1]  **Matt Fredrikson** [1]

## Abstract

We present a black-box model-stealing attack that recovers private vision-tokenizer configurations of deployed vision-language models (VLMs), including the visual patch size and input preprocessing pipeline. The key idea is a task-level side channel induced by ViT-style patchification: when a synthetic grid image is aligned with the hidden patch grid, boundary cues are erased at tokenization, causing periodic accuracy drop. By sweeping the grid cell size and measuring these collapses, we infer the patch size; by introducing padding and a consistency-check test, we further identify whether preprocessing is dynamic- or fixed-resolution and recover the target resize resolution. Across open-source Qwen-VL variants and proprietary models including GPT and Claude, we reliably recover tokenizer-related parameters. Finally, we show that such leakage enables preprocessing-aware transfer attacks and model-targeted adversarial manipulation.

## 1. Introduction

Vision-Language Models (VLMs) deployed via public APIs rely on complex and often undocumented visual preprocessing pipelines. Beyond model weights, these pipelines include architectural and system-level design choices such as vision patch size, input resizing strategies, padding/cropping rules, and target resolutions. These parameters materially affect efficiency, accuracy, and robustness, yet are typically treated as *private deployment-time information* and are not disclosed in APIs or model documentation.

**A "blind spot" in black-box VLMs.** Despite impressive capabilities, black-box VLMs exhibit striking failures on tasks that are trivial for humans. Consider a simple *grid-size counting* query: given an image of a colored $N \times N$ grid

(e.g., Figure 8), ask the model to report $N$. Humans can answer this at a glance. However, we observe that state-of-the-art models (e.g., GPT and Claude) can abruptly collapse on this task for specific cell sizes, producing incorrect counts even when the image remains crisp and unambiguous.

**The "Visual Strawberry" analogy** Just as LLMs struggle to count letters in "strawberry" because BPE tokenization hides individual characters, we show that VLMs fail to count grid cells because **patch tokenization** (patchification) hides visual edges. In both cases, the structural granularity of the tokenizer mismatches the granularity of the information, creating a blind spot.

**Mechanism: Patch-Size Matching (PSM).** Concretely, most VLMs tokenize images via a vision transformer (ViT) that partitions inputs into *non-overlapping square patches*, followed by a patch projection. When salient boundaries (e.g., grid lines) consistently fall *between* patch interiors after preprocessing, boundary cues can be suppressed at the tokenization stage, yielding nearly uniform patch tokens that lack edge information. This effect is not accidental: as we sweep the grid cell size $D$, the alignment condition recurs, causing *periodic accuracy collapses*. We refer to this phenomenon as **Patch-Size Matching (PSM)**: failures occur when the grid frequency becomes commensurate with the patch sampling frequency (e.g., $D = kP$ after preprocessing), creating a repeatable task-level side channel.

**From phenomenon to stealing: recovering private hyperparameters.** This paper studies the following question:

> Can private architectural and preprocessing parameters of black-box VLMs be systematically recovered through API access alone?

We answer this in the affirmative. We present a *black-box model stealing attack* that recovers the *visual patch size* and the *input preprocessing pipeline* of deployed VLMs using only standard image–text queries. Our attack requires no gradients, internal activations, training data, or low-level timing/memory/hardware side channels.

**Stealing patch size and preprocessing under unknown resizing/padding.** If a model preserves input resolution, the

[1]Carnegie Mellon University, Pittsburgh, USA. Correspondence to: Kai Hu <kaihu@cmu.edu>.

*Proceedings of the $43^{rd}$ International Conference on Machine Learning*, Seoul, South Korea. PMLR 306, 2026. Copyright 2026 by the author(s).

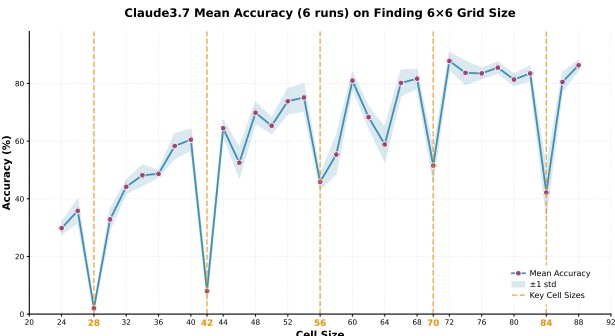

*Figure 1.* Claude3.7 Accuracy on the grid-size counting task ($6 \times 6$) across cell sizes $D$. Periodic drops reveal Patch-Size Matching (PSM) and enable inference of hidden patchification parameters.

period of PSM collapses reveals patch size directly. However, production VLMs often apply undocumented preprocessing (e.g., resizing to fixed targets, dynamic rounding, padding/cropping with unknown anchors), which rescales and shifts the effective patch grid. To handle this, we develop a three-stage black-box attack that: (i) detects whether a model employs *dynamic* or *fixed-resolution* preprocessing, (ii) recovers patch size up to a scaling factor under unknown resizing, and (iii) identifies the target input resolution via a consistency check based on hypothesis testing. Across a range of models, including open-source Qwen-VL variants and proprietary models such as GPT and Claude, our attack reliably recovers patch sizes and preprocessing behaviors.

**Implications for model security.** Extracting architectural and preprocessing parameters has direct security implications. Knowing the preprocessing pipeline improves transfer-based adversarial attacks by avoiding perturbation distortion induced by resizing and padding. Moreover, it enables *model-targeted adversarial attacks*, where a single adversarial image selectively manipulates one VLM while remaining benign to another with a different preprocessing pipeline. **The contributions of this work:**

- **Architectural model stealing:** a black-box attack that extracts visual patch size and preprocessing pipelines from deployed VLMs via standard API queries.

- **Robust methodology:** a three-stage procedure that remains effective under unknown resizing and padding, recovering patch size and target resolution.

- **Downstream security impact:** improved transfer attacks and model-targeted adversarial manipulation enabled by leaked preprocessing parameters.

## 2. Related Work

**Model-stealing (Model-extraction) Attacks** recover functionality or parameters via black-box API access and are a

recognized threat to deployed ML services (Oliynyk et al., 2023; Zhu et al., 2021). Early work demonstrated that "knockoff" models can be trained using labels or probabilities from a target to supervise a surrogate, even across differing architectures (Tramèr et al., 2016; Orekondy et al., 2019). Subsequent research has optimized query efficiency through synthetic data generation and active learning (Kariyappa et al., 2021; Gao et al., 2024), proving practical against diverse real-world APIs (Yu et al., 2020). While defenses like output perturbation and watermarking exist (Kariyappa & Qureshi, 2020), they often face a significant trade-off between utility and robustness (Oliynyk et al., 2023).

**Property Inference and Encoder Stealing.** Another line of work focuses on *property inference*: extracting hyperparameters or architectural details from query responses or side channels like timing (Wang & Gong, 2018; Duddu et al., 2018). Recent studies also highlight the vulnerability of pre-trained encoders, which can be functionally replicated or "fingerprinted" for ownership attribution (Liu et al., 2022; Peng et al., 2022). Our work fits this theme but targets a distinct, security-critical component: the vision tokenizer and its input pipeline. We show that recovering these choices is not just an academic exercise but a prerequisite for model-targeted manipulations.

**Vision Pipelines and Tokenization Robustness.** Modern vision backbones and VLMs (e.g., ViT, CLIP, LLaVA) rely on standardized preprocessing pipelines inherited from ImageNet-style training, involving specific resizing, cropping, and normalization steps (Dosovitskiy et al., 2021; Radford et al., 2021; Liu et al., 2023a). This standardization is a double-edged sword: while it simplifies interoperability, it creates a narrow design space that can be reverse-engineered. Unlike prior work on patch-level vulnerabilities (Liu et al., 2023b), we demonstrate that *patchification itself* leaves a detectable signature in black-box behavior. We exploit these signatures to infer hyperparameters and construct failures targeted to specific model families.

**Black-box VLM Attacks and Jailbreaks.** Attacks on VLMs are increasingly targeting proprietary systems like GPT-4V and Gemini through feature alignment or visual "jailbreaks" designed to elicit policy violations (Dong et al., 2023; Zhao et al., 2023; Tao et al., 2025). We differ in mechanism; our primary goal is to recover the underlying tokenizer and preprocessing properties. We demonstrate that these recovered properties enable more controllable, model-specific manipulations, including systematic benchmark degradation and more effective adversarial transfer.

## 3. The Mechanism: Blindness in Patching

This section explains a deterministic failure mode induced by ViT patchification. When task-relevant boundaries sys-

tematically vanish at the patch-token level, a VLM can become "blind" in a highly structured, *periodic* manner. We formalize a boundary-sensitive grid-size counting probe, analyze when patch tokenization erases boundary cues, and derive the resulting periodic accuracy valleys. We defer the black-box parameter inference procedure to Section 4.

### 3.1. Preliminaries: Grid-Size Counting as a Boundary-Sensitive Probe

**Grid image distribution.** Fix a grid dimension $N$ and a cell size $D$ (in pixels) in the *user image space*. We construct a synthetic image $I(N, D) \in \mathbb{R}^{R \times R \times 3}$ containing an $N \times N$ grid, where $R = ND$. Each cell is filled with a constant RGB color sampled i.i.d. from a small palette. In all experiments we use the binary cube palette $\mathcal{C} = \{0, 255\}^3$ (thus $|\mathcal{C}| = 8$). Let $\mathcal{D}_{N,D}$ denote the resulting distribution over grid images where each of the $N^2$ cell colors is sampled independently from $\mathcal{C}$.

**Difficulty calibration via the palette.** If the palette size $|\mathcal{C}|$ is too large, the task becomes easy via a shortcut (e.g., scanning one row/column and counting distinct colors), which can mask the patch-alignment effect. If $|\mathcal{C}|$ is too small, the grid-size counting task becomes too difficult for the model, and the model tends to guess across all $D$. In practice, we pick $|\mathcal{C}|$ so that the model achieves non-trivial accuracy away from the valley points, while exhibiting sharp collapses near alignment.

**Task and evaluation.** Given an image $I \sim \mathcal{D}_{N,D}$ and a fixed prompt (see Appendix D for the complete prompt), the VLM returns text. We extract a predicted integer $\hat{N}(I)$ via a deterministic parser. For each fixed $(N, D)$, we evaluate on $M$ i.i.d. samples $\{I_m\}_{m=1}^{M}$ from $\mathcal{D}_{N,D}$ and define the empirical accuracy

$$\text{Acc}(N, D) = \frac{1}{M} \sum_{m=1}^{M} \mathbf{1}\{\hat{N}(I_m) = N\}, \qquad (1)$$

which estimates $\text{Pr}_{I \sim \mathcal{D}_{N,D}}[\hat{N}(I) = N]$.

**Patchification.** Most VLMs employ a ViT-like vision encoder that first applies an internal preprocessing pipeline $\pi(\cdot)$ (e.g., resizing/padding/cropping) and then partitions the resulting image into non-overlapping $P \times P$ patches, each mapped to a token embedding. We call $P$ the *patch size* in pixels in the tokenizer input space (after preprocessing).

### 3.2. Patch Size Matching Erases Boundary Cues

**Effective cell size after preprocessing.** Let $D_{\text{eff}}$ denote the effective cell size (in pixels) in the tokenizer input space after preprocessing $\pi(\cdot)$. When $\pi(\cdot)$ approximately performs uniform scaling to a square resolution $S \times S$, $D_{\text{eff}}$

scales linearly with $D$: $D_{\text{eff}} \propto D$. (We make no assumption here about whether $S$ is fixed or dynamic; we only use that $D_{\text{eff}}$ is well-defined for a given input.)

**Patch Size Matching (PSM).** We focus on the alignment regime where the effective cell size is an integer multiple of the patch size: $D_{\text{eff}} = kP, \ k \in \mathbb{Z}^+$. Under this condition, each $P \times P$ patch falls entirely inside a single grid cell. Since each cell interior is constant-color by construction, the pixels within each patch become nearly homogeneous. As a result, the corresponding patch token embedding contains little direct evidence of cell boundaries.

**Why boundaries disappear.** Grid-size counting is fundamentally boundary-based: to infer $N$, a model must detect and aggregate repeated cell separators. When $D_{\text{eff}} = kP$ holds, cell boundaries lie *between* patches rather than inside them, so edge/contrast information is systematically absent from patch tokens. The VLM then lacks the most reliable cue for counting and may fail even though the task is visually obvious to humans.

**Misalignment restores boundary information.** When $D_{\text{eff}}$ is not aligned to $P$, many patches straddle cell boundaries and contain strong local contrast. These boundary-crossing patches inject edge information into token embeddings, making the global grid structure recoverable.

### 3.3. Periodic Accuracy Valleys

**Prediction.** The PSM condition ($D_{\text{eff}} = kP, \quad k \in \mathbb{Z}^+$) holds whenever $D_{\text{eff}}$ hits an integer multiple of $P$. Therefore, as we vary $D$ (and thus vary $D_{\text{eff}}$), we expect the model's accuracy to exhibit sharp local minima whenever

$$D_{\text{eff}} \in \{P, 2P, 3P, \dots\}. \qquad (2)$$

This yields a characteristic *periodic* "valley" pattern in $\text{Acc}(N, D)$. Crucially, the periodicity is induced by the patch-tokenization geometry, not by image content: it arises from whether boundaries are hidden *between* patches.

**Operational implication.** Equation (2) turns patchification into a measurable side channel. If preprocessing preserves a stable mapping between user-space $D$ and tokenizer-space $D_{\text{eff}}$, the valley period reveals $P$. If preprocessing disrupts this mapping (e.g., fixed resizing), the periodicity may be obscured; Section 4 shows how to restore and exploit it in black-box APIs.

## 4. Stealing Architecture Parameters

Building on the mechanism in Section 3, we present a black-box probing attack that infers hidden vision-encoder parameters from structured failure patterns on the grid-size

counting probe. Our goal is to recover: (i) patch size $P$; (ii) whether preprocessing uses *dynamic* or a *fixed* target resolution; and (iii) the fixed target resolution $S$ when applicable.

### 4.1. Threat Model, Unknowns and Assumptions

**Black-box access.** The adversary can submit image–text queries to a deployed VLM and observe its text outputs. No gradients, logits, or internal activations are available.

**Unknown preprocessing.** Let $\pi(\cdot)$ denote the internal image pipeline (resize, padding, cropping, etc.). We distinguish two common classes:

**1. Dynamic resolution:** the model processes inputs at multiple resolutions depending on input size (or chooses from a set) and largely preserves content scale.

**2. Fixed resolution:** the model resizes (and possibly pads/crops) to a fixed $S \times S$ regardless of input.

**Assumptions.** We assume the pre-processing pipeline does not discard content via cropping. This assumption is necessary to define a stable spatial origin (the top-left pixel) for patch-grid counting. Importantly, it can also be verifiable with black-box probes: we place high-contrast fiducial markers (or short OCR strings) within a few pixels of each image boundary and query the model to report their presence/content. Any cropping (e.g., center-crop or content-adaptive crop) would systematically remove or truncate these boundary cues and is thus detectable.

Second, the pipeline treats height and width symmetrically for square inputs, so a square image is mapped to a square image (possibly at a different resolution). With this assumption, we restrict our analysis to square images and avoid additional case analysis over aspect ratios (although it is analyzable using our method).

### 4.2. Stage 1: Unpadded Sweep (Dynamic vs. Fixed)

**Protocol.** Fix $N$ (e.g., $N = 6$) and sweep the user-space cell size $D$ over a range. For each $D$, query $M$ random grid instances and compute $\mathrm{Acc}(N, D)$ using (1). The **decision rule** is given by:

**1. Dynamic-resolution signature.** If $\mathrm{Acc}(N, D)$ shows stable periodic valleys at regular intervals, then $D_{\mathrm{eff}}$ varies proportionally with $D$ across the sweep. The fundamental period directly estimates $P$: $\hat{P} \approx \widehat{\mathrm{period}}(\mathrm{Acc}(N, D))$.

**2. Undetermined signature.** If $\mathrm{Acc}(N, D)$ is smooth/aperiodic (no stable repeating valleys), resizing likely maps many distinct $D$ to non-integer $D_{\mathrm{eff}}$, destroying the alignment pattern. We then proceed to Stage 2 to *restore* periodic observability.

### 4.3. Stage 2: Restoring Periodicity via Zero-Padding

**Anchor-and-pad construction.** Given a grid image of size $R \times R$ where $R = ND$, we embed it into a larger $L \times L$ canvas by zero-padding the bottom and right margins while anchoring content at the top-left. Denote the padded image by $\tilde{I}(N, D; L)$. Figure 8 demonstrates this process.

**Why padding helps under fixed $S$.** Assume the model resizes any $L \times L$ input to $S \times S$ before patchification. Then the entire content is scaled by factor $S/L$, so the effective cell size becomes $D_{\mathrm{eff}} = \dfrac{S}{L} D$. PSM blindness occurs when $D_{\mathrm{eff}} = kP$, implying valleys at user-space cell sizes $D = k \cdot \dfrac{PL}{S}$, $k \in \mathbb{Z}^+$. Thus, for each fixed canvas size $L$, the observed valley period in $D$ is

$$T(L) = \frac{PL}{S} \quad \Longrightarrow \quad \frac{S}{P} = \frac{L}{T(L)}. \tag{3}$$

**Protocol.** Choose a set of canvas sizes $\mathcal{L}$ (e.g., common resolutions or a dense range). For each $L \in \mathcal{L}$, sweep $D$ over feasible values ($ND \leq L$), estimate the dominant period $T(L)$ from repeating valleys, and compute an estimate of $S/P$ via (3). Intersecting constraints across multiple $L$ yields a small candidate set of $(S, P)$.

### 4.4. Hypothesis Testing for Periodic Drops

To avoid *subjective* conclusions about periodic accuracy collapses (e.g., in Figure 1), we use a statistically principled, nonparametric hypothesis test to (i) detect whether a periodic drop exists and (ii) estimate its period.

**Setup.** Let $\{x_i\}_{i=1}^n$ be the tested cell sizes (the $x$-axis) and $\{y_i\}_{i=1}^n$ the corresponding accuracies. For a candidate period $T$, we partition the accuracies into two groups:

$$\begin{aligned}
G_0(T) &= \{y_i : x_i \bmod T = 0\}, \\
G_1(T) &= \{y_i : x_i \bmod T \neq 0\},
\end{aligned} \tag{4}$$

with sizes $n_0(T), n_1(T)$ and sample means/variances $(\bar{y}_0(T), s_0^2(T))$ and $(\bar{y}_1(T), s_1^2(T))$.

**Test statistic.** Under a periodic drop at multiples of $T$, we expect $\bar{y}_0(T) < \bar{y}_1(T)$. We quantify this using a one-sided Welch-style standardized mean difference:

$$t(T) = \frac{\bar{y}_0(T) - \bar{y}_1(T)}{\sqrt{s_0^2(T)/n_0(T) + s_1^2(T)/n_1(T)}}. \tag{5}$$

Smaller $t(T)$ indicates a stronger drop at multiples of $T$.

**Permutation $p$-value (distribution-free).** To avoid parametric assumptions on $\{y_i\}$, we estimate significance by

permutation. Let $\pi$ be a random permutation of $\{1, \ldots, n\}$, and define the permuted accuracies $y_i^{(\pi)} = y_{\pi(i)}$ while keeping $\{x_i\}$ fixed. We recompute $t^{(\pi)}(T)$ from $\{(x_i, y_i^{(\pi)})\}$. With $B$ permutations, the one-sided permutation $p$-value is

$$p(T) = \frac{1 + \sum_b \mathbf{1}\left[t^{(b)}(T) \le t(T)\right]}{B + 1}. \quad (6)$$

We select $\hat{T} = \arg\min_{T \in \mathcal{T}} p(T)$. We will show that the observed $T$ leads to a significantly small $p$-value while other candidates lead to much greater p-value. This procedure provides an objective criterion for period identification without relying on visual inspection.

### 4.5. Stage 3: Identifying the True Target Resolution via Relative Answer Consistency

Stage 2 yields a small candidate set of fixed target resolutions $\{S_j\}$ (and corresponding $P_j$ via (3)). We infer the true resolution from the candidate set. **Intuition**: if the true internal target is $S$, then querying an image at an arbitrary size and querying the same content pre-resized to $S$ should be more self-consistent than pre-resizing to a wrong candidate $S' \ne S$ (which introduces an extra resampling step).

**Paired evaluation on the same images.** Let $f(\cdot)$ be the black-box VLM API and $\mathrm{parse}(\cdot)$ extract the predicted integer $\hat{N}$ from the text output. Let $S_a$ and $S_b$ denote two resolution candidates. For each test image $x$, we obtain:

$$
\begin{aligned}
a_0(x) &:= \mathrm{parse}(f(x)), \\
a_{S_a}(x) &:= \mathrm{parse}(f(\mathrm{resize}(x, S_a))), \\
a_{S_b}(x) &:= \mathrm{parse}(f(\mathrm{resize}(x, S_b))). \quad (7)
\end{aligned}
$$

We compare $S_a$ and $S_b$ by checking which pre-resize produces an answer that matches the baseline $a_0(x)$ more often on the *same* images. The winner is more likely to be the true target resolution:

**Win counts (non-tie outcomes only).** Define the non-tie win indicators:

$$
\begin{aligned}
\mathrm{win}_a(x) &= \mathbf{1}\{a_{S_a}(x) = a_0(x) \ \wedge \ a_{S_b}(x) \ne a_0(x)\}, \\
\mathrm{win}_b(x) &= \mathbf{1}\{a_{S_b}(x) = a_0(x) \ \wedge \ a_{S_a}(x) \ne a_0(x)\}. \quad (8)
\end{aligned}
$$

Over a test set $\mathcal{X} = \{x^{(t)}\}_{t=1}^T$, let

$$Y_a = \sum_{t=1}^T \mathrm{win}_a(x^{(t)}), \qquad Y_b = \sum_{t=1}^T \mathrm{win}_b(x^{(t)}). \quad (9)$$

We ignore tie cases where both match (or both mismatch) the baseline, since they provide no discriminative evidence between $S_a$ and $S_b$.

**Significance test.** Conditioned on non-tie trials, under the null hypothesis that $S_a$ and $S_b$ are equally consistent with the baseline, the wins are symmetric. We compute

$$z = \frac{Y_a - Y_b}{\sqrt{Y_a + Y_b}}, \quad (10)$$

and select $S_a$ over $S_b$ if $z > \Phi^{-1}(1 - \alpha)$ (and vice versa if $z < -\Phi^{-1}(1 - \alpha)$), with $\alpha = 10^{-3}$.

**Tournament selection across candidates.** We perform the above pairwise comparison in a tournament over all candidates $\{S_j\}$ (e.g., bracketed elimination or round-robin with multiple comparisons control), and output the unique $\hat{S}$ that consistently wins against alternatives. Finally, $\hat{P}$ is obtained from the Stage-2 ratio estimate: $\hat{P} \approx \dfrac{\hat{S}}{\widehat{S/P}}$.

## 5. Attack Results

We evaluate the three-stage probing pipeline in Section 4 as a black-box *inference procedure* that recovers: (i) whether the VLM uses dynamic-resolution or fixed-resolution preprocessing, (ii) the visual patch size $P$, and (iii) for fixed-resolution pipelines, the target resize resolution $S$. Unless otherwise stated, we use the $N \times N$ grid-size counting task with $N=6$, sweep cell sizes $D$, and report $\mathrm{Acc}(N, D)$. We report additional results with other $N$ in Appendix B.

**Randomness and statistical significance.** We observe two sources of randomness: (i) closed-source VLM APIs can be non-deterministic even at temperature 0, and (ii) our grid probes vary due to i.i.d. color sampling. To address this, we set $M = 200$ to compute Equation 1 and repeat the experiments 6 times. All figures are presented as mean values with standard deviation error bars.

### 5.1. Stage 1: Unpadded periodicity diagnoses dynamic-resolution and reveals $P$

**Protocol and inference rule.** We query unpadded grid images of resolution $R = ND$ and examine $\mathrm{Acc}(6, D)$ as a function of $D$. Under a dynamic-resolution pipeline, patchification operates on (or near) the input resolution, so grid-cell boundaries periodically align with patch boundaries, inducing stable accuracy minima at $D \approx kP$. In contrast, fixed-resolution resizing typically smooths or suppresses this periodic signature. We therefore treat *repeated, stable* minima over a broad $D$ range as evidence of dynamic-resolution and estimate $P$ as the fundamental period.

**Open-source Qwen Models.** Figure 2 (Qwen3-VL 30B) exhibits clear, repeated minima at $D \in \{48, 64, 80\}$, implying a fundamental period $P=16$. Appendix B provides

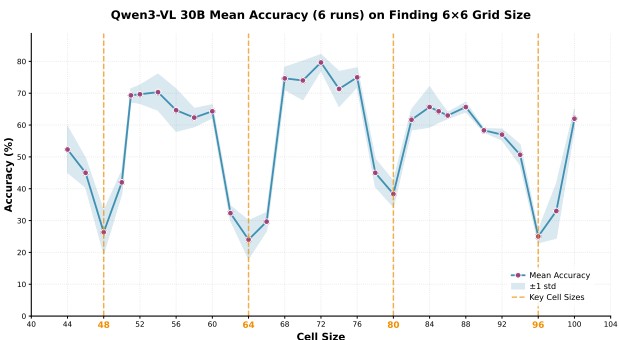

*Figure 2.* Qwen3-VL 30B accuracy on the grid-size counting task $(6 \times 6)$ across cell sizes $D$ (no padding).

results for the Qwen2.5-VL model. Since Qwen models are open-source, these inferences can be directly verified.

**Claude Models** Figure 1 shows pronounced periodic minima aligned with multiples of 14 (e.g., $D \in \{k \cdot 14 : k \in \{2, 3, 4, 5, 6\}\}$), supporting dynamic-resolution preprocessing and implying $P=14$. To quantitatively show this period, we conduct the hypothesis test in Section 4.4. We consider period candidates $\mathcal{T} = \{8, 9, \cdots, 27\}$, and report the Welch statistic and p-value. Table 1 shows the results. Due to space limit, we only list the best candidates. We can see $T = 14$ is the most significant period ($T = 21$ is also significant because $21 = 14 \times 1.5$). Appendix B provides results for Claude 4.5. Although this model has stronger reasoning capability, it still has the same blind spot.

*Table 1.* Statistic and p-value of period candidates for the Claude 3.7 curve. Smaller $t(T)$ and p-values are better.

| T | 14 | 21 | 23 | 15 | 12 | 10 |
|---|-----|------|------|-------|-------|------|
| t(T) | -3.5 | -2.87 | -2.50 | -0.28 | -0.11 | 0.00 |
| p-value | 1e-4 | 2e-3 | 0.06 | 0.33 | 0.39 | 0.49 |

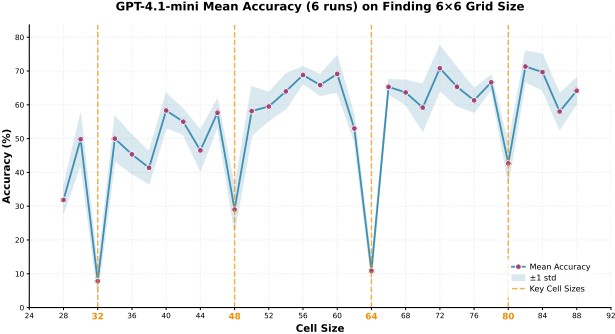

*Figure 3.* GPT-4.1-mini accuracy on the grid-size counting task $(6 \times 6)$ across cell sizes $D$ (no padding).

**GPT-4.1-mini.** Figure 3 shows stable minima at $D \in \{k \cdot 16 : k \in \{2, 3, 4, 5\}\}$, implying dynamic-resolution

preprocessing and $P=16$.

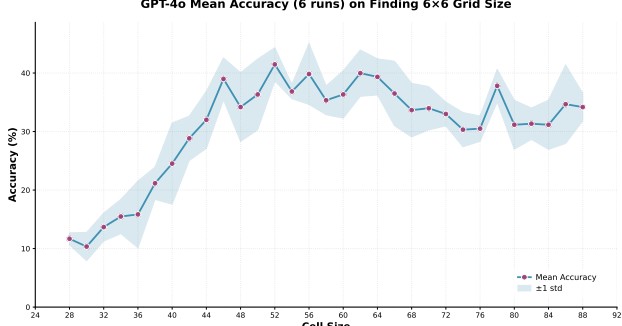

*Figure 4.* GPT-4o accuracy on the grid-size counting task $(6 \times 6)$ across cell sizes $D$ (no padding).

**GPT-4o: absence of unpadded periodicity suggests fixed-resolution.** Figure 4 is comparatively smooth and does not exhibit stable, repeating minima across $D$. This behavior is consistent with fixed-resolution preprocessing (e.g., resizing all inputs to a target size $S$ before patchification). However, Stage 1 alone cannot rule out alternative explanations (e.g., aggressive anti-aliasing). Stages 2 & 3 reintroduce a controllable alignment signal to disambiguate these cases.

### 5.2. Stage 2: Padding restores periodicity and estimates the scale ratio $S/P$

**Protocol.** For suspected fixed-resolution models, we embed the $R \times R$ grid into a larger $L \times L$ canvas via zero padding (grid anchored at the top-left), then sweep $D$. If the model resizes to a fixed target $S$, then changing $L$ induces a predictable rescaling of the effective cell size, and periodic minima re-emerge with a period that scales with $L$.

**GPT-4o results and scale-ratio inference.** At $L=448$ (top-left of Figure 5), we observe clear minima at $D \in \{28, 42, 56, 70\}$ with period $T(448) = 14$. The Stage 2 analysis implies the scale ratio

$$\frac{S}{P} \approx \frac{L}{T(L)} = \frac{448}{14} = 32. \tag{11}$$

With $\frac{S}{P} = 32$, we predict $T(L) = L/32$, i.e., $T \in \{15, 16, 17\}$ for $L \in \{480, 512, 544\}$, respectively. The remaining three panels match this prediction. We occasionally observe isolated dips that do not form a period. We treat these as outliers due to stochasticity or model idiosyncrasies. Appendix C provides hypothesis testing for visually inspected periods of GPT models (Figure 3 and Figure 5).

**Candidate set.** The cross-$L$ linear scaling strongly supports a fixed-resolution resize to a single $S$. Using Eq. (11),

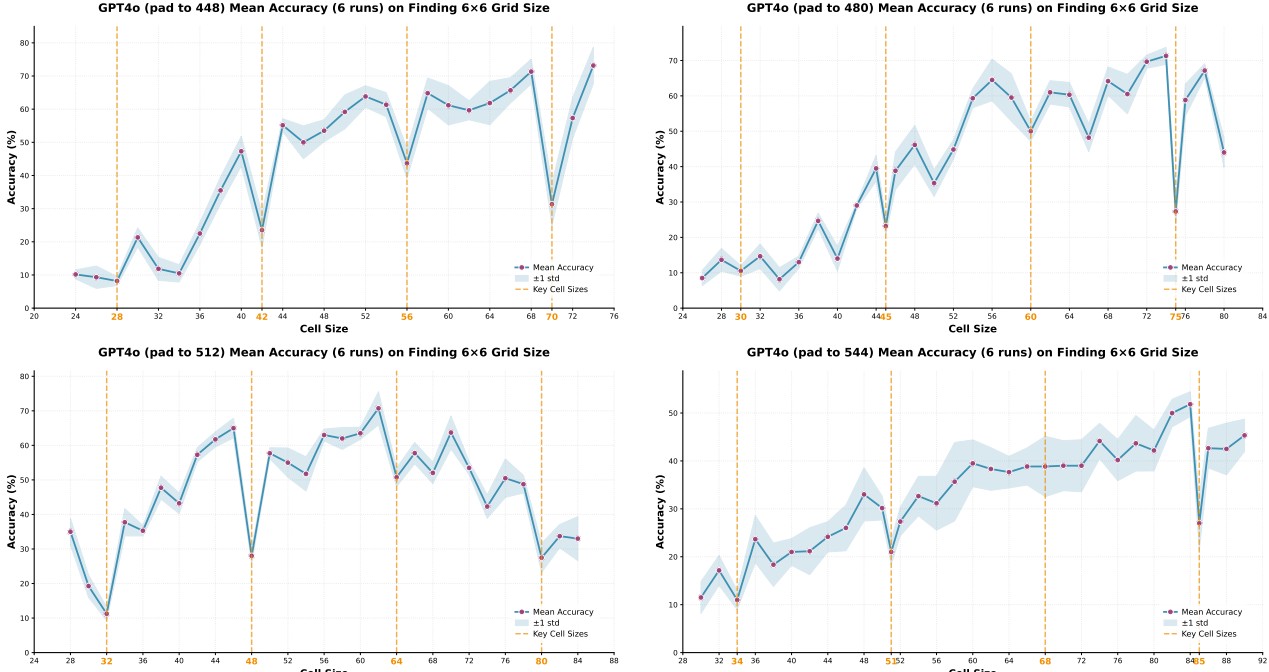

*Figure 5.* GPT-4o accuracy on the grid-size counting task ($6 \times 6$) under zero padding to different canvas sizes $L$. Top-left: $L$=448, top-right: $L$=480, bottom-left: $L$=512, bottom-right: $L$=544.

we form a small ambiguity set of candidates:

$$\{(L_j, T_j)\} = \{(448, 14), (480, 15), (512, 16), (544, 17)\}. \quad (12)$$

where each $L_j$ is a plausible $S$ (implying $P = S/32$). We prioritize common configurations unless Stage 3 fails to distinguish them.

**GPT's Dynamic input pipeline**   The GPT API provides an option named "detail" to specify image quality, with available values, including "low", "high", and "auto". According to OpenAI documents, "detail=high" adds additional sub-crops of the input images as model input and "detail=auto" will automatically determine which to use between "low" and "high". Unless otherwise stated, we use "detail=low" by default to reduce token use. However, it is vital to verify if the attack pattern would remain valid under this multi-scale processing. Figure 6 and Figure 7 show GPT-4o accuracy under zero padding to 448 for "detail=high" and "detail=auto" respectively. Figure 13 shows GPT-4o accuracy under zero padding to different canvas size.

**Open-source InternVL 3.5 30B model** (Wang et al., 2025) adopts similar pre-processing as GPT-4o that resizes the input images to an input size of 448. We provide additional verification on this model. Appendix B provide the analysis in Figure 14.

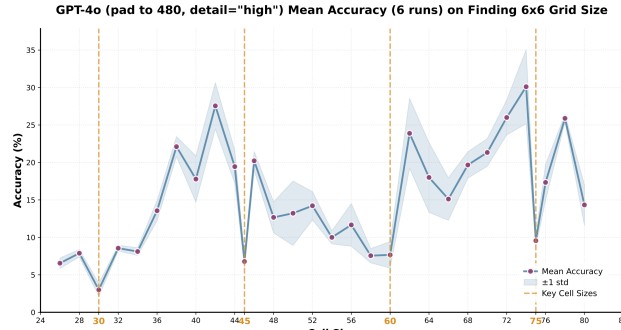

*Figure 6.* GPT-4o accuracy under zero padding to 480 for "detail=high"

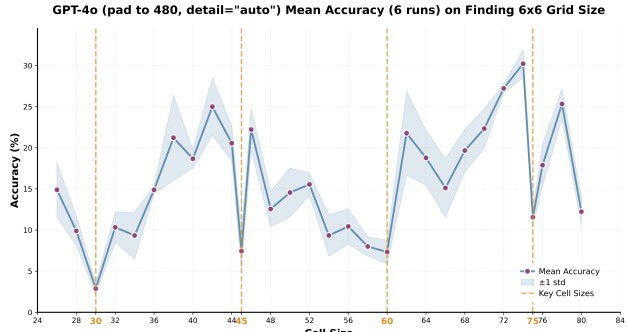

*Figure 7.* GPT-4o accuracy under zero padding to 480 for "detail=auto"

## 5.3. Stage 3: Consistency check selects the true $(S, P)$

Table 2 implements the consistency test in Section 4.5. We compare candidates pairwise and compute a one-sided $z$-statistic. It shows that $L=512$ wins against all other candidates with overwhelming significance ($z \geq 10$ in every comparison involving 512). In contrast, comparisons among $\{448, 480, 544\}$ yield much smaller effects. Using a significance level $\alpha = 10^{-3}$ (threshold $\Phi^{-1}(1 - \alpha) \approx 3.1$), only $L=512$ is consistently significant against all alternatives. We therefore infer **a fixed target resolution** $S=512$ for GPT-4o. Combining with Eq. (11) yields $P=16$.

*Table 2.* Consistency check for GPT-4o over four target-resolution candidates $L \in \{448, 480, 512, 544\}$. $Y_i$ ($Y_j$) counts non-tie wins for $L_i$ ($L_j$). The $z$-statistic follows Section 4.5.

| $L_i$ | $L_j$ | $Y_i$ | $Y_j$ | Ties | $z$ | Winner |
|-------|-------|-------|-------|------|------|--------|
| 512 | 448 | 320 | 109 | 771 | 10.2 | 512 |
| 512 | 480 | 383 | 118 | 700 | 11.8 | 512 |
| 512 | 544 | 413 | 125 | 662 | 13.3 | 512 |
| 480 | 544 | 202 | 178 | 820 | 1.2 | 480 |
| 480 | 448 | 207 | 183 | 810 | 1.2 | 480 |
| 544 | 448 | 193 | 172 | 835 | 1.1 | 544 |

**Scope and limitations.** Our inference holds under the threat model in Section 4.1. If a system uses content-adaptive crops, multi-scale routing, or stochastic resizing such that no single fixed $S$ exists, Stage 2 may fail to stabilize a period and Stage 3 may not identify a dominant winner; in that case the correct output is "non-identifiable under our threat model" rather than a forced point estimate.

**Attack Cost.** For a fixed configuration, we sweep roughly 40 candidate cell sizes and issue 1,200 image queries per cell size, totaling $4.8 \times 10^4$ queries. For non-reasoning models, this corresponds to on the order of $10^7$ input/output tokens in aggregate. We emphasize that the 1,200-image setting is chosen to obtain tight confidence intervals. In practice, the same phenomenon is already visible with about 200 images per cell size. Under typical API pricing, this smaller run costs roughly \$10~\$50 depending on the provider and the response length.

## 6. Model-Targeted Adversarial Attacks

In this section, we demonstrate that the architectural parameters recovered in Section 5, specifically the exact input resolution $S$ and patch size $P$, are not merely theoretical curiosities but critical security vulnerabilities.

**Motivation: The Preprocessing Gap.** Standard adversarial attacks typically assume a generic preprocessing pipeline (e.g., standard $224 \times 224$ resizing). However, as our stealing attack revealed, production models like GPT-4o and Claude 3.7 employ vastly different pipelines ($\pi_{GPT}$ vs. $\pi_{Claude}$). Ignorance of these pipelines leads to perturbation distortion (aliasing) during resizing. Conversely, **knowledge** of these pipelines enables us to craft **Model-Targeted Attacks**: adversarial examples that are lethal to a specific target model while remaining benign to others.

### 6.1. Methodology

Let $x$ be the clean image and $\delta$ be the perturbation with $\|\delta\|_\infty \leq \epsilon$. We utilize an ensemble of $N$ white-box surrogate models $\{F_i\}_{i=1}^N$. Let $\ell_i(x)$ denote the attack loss.

**Baseline (Model-Agnostic) Transfer Attack.** The standard approach optimizes $\delta$ to fool the surrogates on average. Random data augmentation $a \sim A$ (e.g., random crop, random horizontal flip) is often applied to improve transferability. The objective is:

$$\delta^* = \operatorname*{argmin}_{\|\delta\|_\infty \leq \varepsilon} \sum_{i=1}^N \mathbb{E}_{a \sim A} \ell_i(F_i(a(x + \delta))). \quad (13)$$

This perturbation often transfers poorly or unpredictably because the victim's resizing operation distorts the high-frequency adversarial patterns.

**Model-Targeted Attack (Ours).** We leverage the stolen pipelines $\pi_A$ (Target, e.g., GPT-4o's fixed 512) and $\pi_B$ (Non-Target, e.g., Claude 3.7's dynamic). We optimize a **contrastive objective**:

$$\delta^* = \arg \min_{\|\delta\|_\infty \leq \varepsilon} \sum_{i=1}^N \mathbb{E}_{a \sim A} \Big[ \ell_i\big(F_i(a \circ \pi_A(x + \delta))\big) \\ - \alpha\, \ell_i\big(F_i(a \circ \pi_B(x + \delta))\big) \Big]. \quad (14)$$

Here, the first term minimizes the attack loss under the target's pipeline (ensuring attack success). The second term *maximizes* the attack loss under the non-target's pipeline (ensuring the attack fails, i.e., the image remains benign or non-targeted). $\alpha$ is a hyperparameter balancing the two objectives (set to 0.5). Physically, this optimization hides the adversarial signal in frequencies that are preserved by $\pi_A$ but destroyed or aliased by $\pi_B$.

### 6.2. Empirical Verification

**Setup.** We evaluate on the **NIPS 2017 Adversarial Challenge** dataset (K et al., 2017) (1,000 images). Each image has a source class and a target class. We target two commercial SOTA models with distinct architectures identified by our stealing attack: **GPT-4o** (Fixed Resolution $512 \times 512$, $P = 16$) and **Claude 3.7** (Dynamic Resolution, $P = 14$).

*Table 3.* **Model-Targeted Attack Results.** ASR (%) on the NIPS 2017 dataset. "Targeted GPT" aims to attack GPT-4o while sparing Claude, and vice versa. ↑ indicates the intended higher ASR on the target and ↓ indicates the intended lower ASR on the non-target.

| Attack Method | GPT-4o | Claude 3.7 |
|---|---|---|
| SSA-CWA (Dong et al., 2023) | 42.3 | 9.1 |
| AnyAttack (Zhang et al., 2025) | 36.3 | 7.8 |
| Baseline (Eq. 13) | 78.5 | 13.5 |
| Targeted to GPT-4o | 86.5(↑) | 4.5 (↓) |
| Targeted to Claude 3.7 | 62.5(↓) | 18.5 (↑) |

We use three surrogate models: ViT-H (DNF) (Fang et al., 2023), ViT-SO400M (SigLIP) (Zhai et al., 2023), and ViT-H (DataComp-1B) (Gadre et al., 2023). The attack budget is $\varepsilon = 8/255$. We report the **Attack Success Rate (ASR)**, defined as the percentage of images where the victim model outputs the target class. For a successful targeted attack, we expect high ASR on the target and low ASR on the non-target. Appendix D provides more implementation details.

**Results.** Table 3 presents the results. The Baseline attack achieves high ASR on GPT-4o ($78.5\%$) but fails to distinguish between models. In contrast, our **Targeted GPT-4o** attack increases ASR on GPT-4o to **86.5**% (+8.0%) while successfully suppressing ASR on Claude 3.7 to **4.5**% (-9.0%). Similarly, the **Targeted Claude** attack flips the dominance, achieving 18.5% on Claude (a relative 37% improvement over baseline) while reducing GPT-4o's ASR.

This confirms that knowledge of the preprocessing pipeline enables precise, surgical strikes against black-box VLMs.

**Limitations.** The effectiveness of this targeted attack relies on the existence of meaningful differences between $\pi_A$ and $\pi_B$. If two models share identical preprocessing (e.g., both use fixed $336 \times 336$), distinguishing them via resolution-based targeting would be infeasible. However, given the diversity in current commercial deployments , this vector remains highly relevant.

## 7. Conclusion

We identify and exploit a task-level side channel induced by ViT patchification in deployed VLMs. By sweeping a human-trivial grid-size counting probe, we observe periodic accuracy valleys that reveal patch-alignment failures and encode the hidden patch size. We further develop a three-stage black-box pipeline that remains effective under unknown resizing and padding, and can also pinpoint a fixed target resolution via a consistency check. Across open-source and proprietary models, our attack recovers tokenizer-related deployment parameters (e.g., $P$, dynamic vs. fixed prepro-

cessing, and $S$ when applicable). Beyond parameter leakage, we show that knowing preprocessing enables stronger pipeline-aware transfer attacks and even model-targeted manipulations. These results suggest that "implementation details" in the vision frontend are security-relevant secrets, motivating mitigations such as randomized/stochastic preprocessing, boundary-preserving tokenization, and query controls for probing-style extraction.

## Acknowledgment

This research is based upon work supported in part by the Office of the Director of National Intelligence (ODNI), Intelligence Advanced Research Projects Activity (IARPA), via 560000C260017. The views and conclusions contained herein are those of the authors and should not be interpreted as necessarily representing the official policies, either expressed or implied, of ODNI, IARPA, or the U.S. Government. The U.S. Government is authorized to reproduce and distribute reprints for governmental purposes notwithstanding any copyright annotation therein.

## Impact Statement

Our work exposes a previously underappreciated task-level side channel in deployed vision-language models: ViT-style patchification and undocumented preprocessing can systematically erase boundary cues under patch-aligned inputs, enabling a black-box adversary to infer private deployment-time configuration such as patch size, whether preprocessing is dynamic vs. fixed-resolution, and (when applicable) the target resize resolution—information that can materially improve downstream attacks (e.g., preprocessing-aware transfer attacks and model-targeted adversarial manipulation).

Although our proposed attack method is effective, it can be mitigated. Random or routing test-time preprocessing and using convolutional layers for image patch division can protect the hidden hyperparameters from being stolen. More importantly, our work provides a new perspective on the threats faced by deployed vision-language models, quantifies the degree of information loss caused by tokenization, and offers insights into corresponding mitigation measures.

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

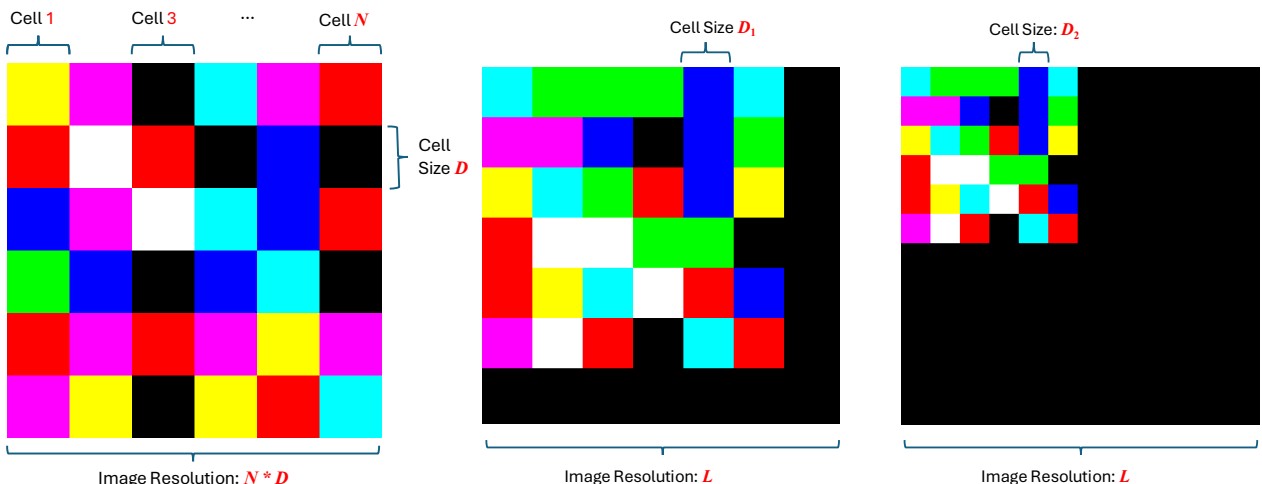

*Figure 8.* Grid size counting: an $N \times N$ grid with $D \times D$ pixel cells. The task is to ask VLMs to find the grid size $N$.

## A. Further Demonstration of Color Grid Image

The left image of Figure 8 provides a visual example of the color grid image we use to prompt black-box VLM APIs in the first stage. The middle and right two images of Figure 8 show the color grid image used in Stage 2. Two $N \times N$ grid images with different cell sizes $D_1, D_2$ are zero-padded to size $L$, where $L \geq N \cdot \max(D_1, D_2)$. Padding is applied from the bottom-right so that content remains anchored at the top-left, because visual transformers read visual tokens from top-left. Algorithm 1 provides a pseudocode to generate the grid images with and without padding.

**Algorithm 1** Grid Image Generation in PyTorch-like Code

```
# D: cell size
# N: number of grids
# L: Zero pad the image to this size if > C * N.
# M: number of generated images.
# seed: A random seed to generate the same color palettes for different cell size C.

def generate_grids(N, D, L=-1, M=1000, seed=42):
  set_seed(seed)
  grids = random_choice([0, 255], size=(M, 3, N, N))
  images = interpolate(grids, scale_factor=D, interpolate_mode="nearest")

  size = N * D
  if L > size:
    background = zeros(M, 3, L, L)
    background[:, :, :size, :size] = images
    images = background

  return images
```

## B. Results on More VLMs

**Qwen2.5 VL** Figure 9 (Qwen2.5-VL 72B) has lower overall accuracy and larger collapse regions, but the minima are most consistent with a period near 14 (e.g., around $D \in \{42, 56, 70, 84, 98\}$), yielding $P=14$.

**Claude 4.5** Figure 10 shows stable minima at $D \in \{k \cdot 14 : k \in \{2, 3, 4, 5, 6\}\}$, implying dynamic-resolution preprocessing and $P=14$.

**Results with different $N$** Figure 11 and Figure 12 present the results of GPT-4.1-mini for finding $5 \times 5$ and $7 \times 7$ grids respectively. It shows that our probing is robust to the choices of $N$.

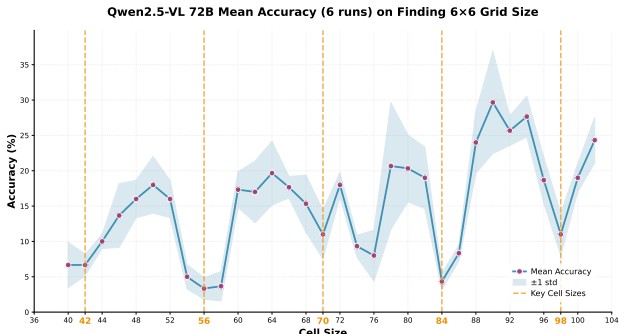

*Figure 9.* Qwen2.5-VL 72B accuracy on the grid-size counting task ($6 \times 6$) across cell sizes $D$ (no padding).

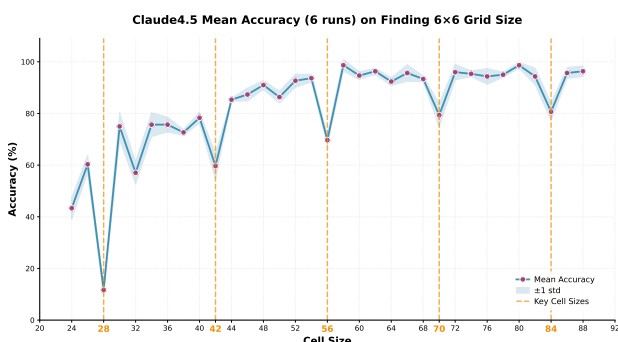

*Figure 10.* Claude 4.5 accuracy on the grid-size counting task ($6 \times 6$) across cell sizes $D$ (no padding).

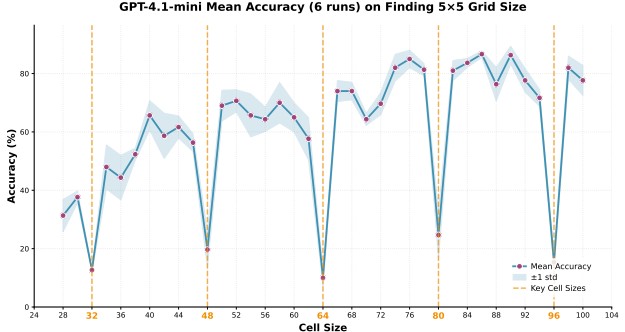

*Figure 11.* GPT-4.1-mini accuracy on the grid-size counting task ($5 \times 5$) across cell sizes $D$ (no padding).

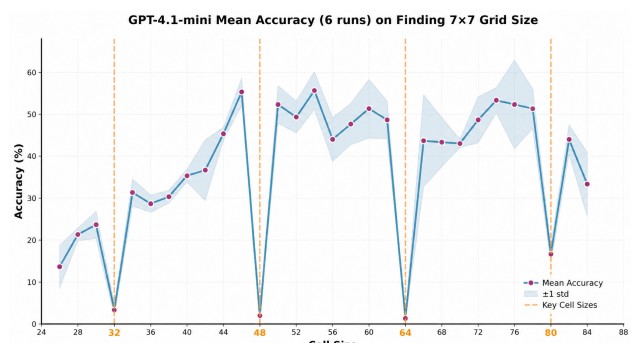

*Figure 12.* GPT-4.1-mini accuracy on the grid-size counting task ($7 \times 7$) across cell sizes $D$ (no padding).

**GPT-4o results under different resizing pipeline** The left three figures in Figure 13 shows GPT-4o accuracy under zero padding to different canvas size with "detail=high". The right three figures in Figure 13 shows GPT-4o accuracy under zero padding to different canvas size with "detail=auto".

**InternVL 3.5 results** We provide additional results for InternVL 3.5 30B in Figure 14. When under no padding (left figure in Figure 14), there is no periodic pattern. When under zero padding to 448 and 512, the accuracy curves show a periodic drop with a period of 14 and 16. In our consistency-check attacks, the S=448 candidate showed a significantly higher winning rate compared to other candidates (e.g., 512), providing a second, independent layer of verification.

**Summary of Model Steal Results** Table 4 summarizes the model steal results.

| Model | Inferred Resize | Inferred $P$ | Inferred $S$ |
|---|---|---|---|
| Qwen3-VL 30B | Dynamic-resolution | 16 | – |
| Qwen2.5-VL 72B | Dynamic-resolution | 14 | – |
| Claude 3.7 & 4.5 | Dynamic-resolution | 14 | – |
| GPT-4.1-mini | Dynamic-resolution | 16 | – |
| GPT-4o | Fixed-resolution | 16 | 512 |

*Table 4.* For dynamic-resolution pipelines, $S$ is not a single fixed value.

## C. Hypothesis Testing for Periodic Accuracy Valleys

Table 5 shows the hypothesis testing results for Periodic Accuracy Valleys. The Welch-style statistics (Equation 5) and the corresponding p-value (Equation 6) are reported. We only list the candidates with the most significant statistics. We can see the most significant period for GPT-4.1-mini and GPT-4o pad to 448, 480, 512, 544 are 16, 14, 15, 16, 17. These results

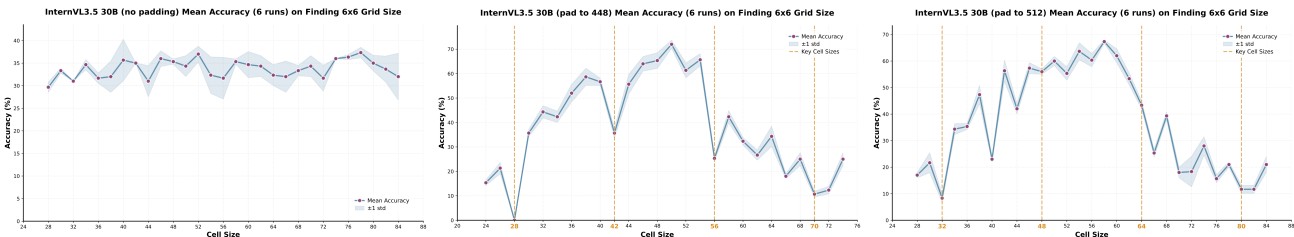

*Figure 13.* GPT-4o accuracy on the grid-size counting task ($6 \times 6$) under zero padding to different canvas sizes $L = 448, 480, 512$ "detail=high" (left) and "detail=auto" (right) respectively.

*Figure 14.* InternVL accuracy on the grid-size counting task under no padding, zero padding to 448 and zero padding to 512.

match the visual inspection periods shown in Figure 3 and Figure 5.

*Table 5.* Statistic and p-value of period candidates for the GPT models. Smaller $t(T)$ and p-values are better.

| | T | 16 | 8 | 19 | 24 | 17 | 20 | 12 | 14 | 10 | 15 |
|---|---|---|---|---|---|---|---|---|---|---|---|
| GPT-4.1-mini | t(T) | -4.97 | -1.57 | -0.40 | -0.30 | 0.49 | 0.40 | 0.38 | 0.47 | 0.52 | 0.76 |
| | p-value | 1e-4 | 0.01 | 0.24 | 0.28 | 0.70 | 0.70 | 0.76 | 0.83 | 0.89 | 0.89 |
| | T | 14 | 21 | 8 | 12 | 13 | 24 | 26 | 16 | 15 | 17 |
| GPT-4o pad to 448 | t(T) | -2.53 | -4.82 | -0.42 | -0.32 | -0.38 | -0.28 | -0.38 | -0.10 | -0.17 | -0.13 |
| | p-value | 1e-4 | 1e-4 | 0.18 | 0.28 | 0.32 | 0.32 | 0.32 | 0.41 | 0.41 | 0.45 |
| | T | 15 | 10 | 20 | 17 | 26 | 13 | 9 | 16 | 19 | 14 |
| GPT-4o pad to 480 | t(T) | -1.66 | -0.71 | -0.47 | -0.18 | -0.05 | -0.05 | 0.02 | 0.03 | 0.05 | 0.08 |
| | p-value | 1e-4 | 0.07 | 0.24 | 0.42 | 0.46 | 0.46 | 0.51 | 0.52 | 0.53 | 0.56 |
| | T | 16 | 8 | 12 | 17 | 15 | 24 | 20 | 18 | 9 | 10 |
| GPT-4o pad to 512 | t(T) | -2.37 | -1.40 | -0.75 | -0.77 | -0.38 | -0.38 | -0.21 | -0.17 | -0.17 | 0.00 |
| | p-value | 1e-4 | 0.01 | 0.09 | 0.17 | 0.29 | 0.29 | 0.36 | 0.37 | 0.37 | 0.47 |
| | T | 17 | 19 | 16 | 15 | 10 | 8 | 21 | 20 | 13 | 26 |
| GPT-4o pad to 544 | t(T) | -1.57 | -0.35 | -0.08 | -0.08 | -0.06 | 0.02 | 0.25 | 0.21 | 0.33 | 0.33 |
| | p-value | 0.01 | 0.30 | 0.44 | 0.44 | 0.44 | 0.52 | 0.62 | 0.62 | 0.66 | 0.66 |

# D. Implementation Details

We use the following prompt to query black-box VLMs for the grid-counting task:

```
There is an N–by–N grid in the image. Each grid cell is filled with a random color.
Observe the grid carefully and find its grid size.
```

## D.1. Implementation Details of Model Targeted Attack

We follow the settings in (Hu et al., 2025) to conduct the adversarial attack.

**Optimization** We apply random crop, random horizontal flip, drop path and patch drop as the data augmentation. The adversarial example is initialized as 0, optimized with Adam optimizer using a learning rate of $\frac{1}{255}$ for 1,000 steps.

**Evaluation** we use the following template to prompt the victim VLM to generate a caption for the image:

```
Provide a detailed description of the image using no more than five sentences.
```

Next we use the GPT-4.1 judge to evaluate if the caption corresponds to the ground truth category, the target category, neither or both. We use the following template to prompt GPT-4.1. An attack is considered successful only if GPT-4.1 responds with "B".

```
The paragraph is a description of an image:
{{caption}}

Which of the following best describes the category of the object in the image:
A) {{ground truth category}}.
B) {{targeted category}}.
C) both A and B.
D) neither A nor B.
Answer with "A)", "B)", "C", or "D)".
```

