# OpenReview forum: "Steal the Patch Size: Adversarially Manipulate Vision Language Models"
_ICML.cc/2026/Conference — ICML 2026 regular_

### Official Review · Reviewer_6Eye · 2026-03-11

**Soundness:** 2
**Presentation:** 2
**Significance:** 3
**Originality:** 3
**Overall Recommendation:** 4
**Confidence:** 3

**Summary:**

This paper proposed a black-box model-stealing attack against vision-language models using only API queries, which recovers configurations in the input preprocessing pipeline including visual patch size and image resolution. The authors first observed a phenomenon, termed Patch-Size Matching, where model accuracy on a synthetic grid-counting task drops periodically when the post-preprocessing grid cell size becomes parallel with the hidden ViT patch grid, causing boundary cues to disappear during tokenization. The authors then leveraged this observation to build a three-stage attack that uses grid-size sweeps, padding-based restoration of periodicity, and a consistency check to infer whether preprocessing is dynamic or fixed resolution, recover the patch size, and identify the target resize resolution.

**Compliance With Llm Reviewing Policy:**

Affirmed.

**Final Justification:**

I have no further questions. And I have raised my score.

**Key Questions For Authors:**

Can authors provide more verifiable evaluations on VLM to check the efficacy of the method? How likely and stable the method is to extract the exact patch size and resolution? and provide solid evidences supporting authors' Blindness in Patching observation?

**Limitations:**

Yes

**Strengths And Weaknesses:**

Strengths:
1. The paper presents a security-relevant black-box side channel for VLMs, using patch geometry to infer private preprocessing properties such as patch size and resize strategy.
2. The technical intuition is clear, and the proposed multi-stage attack pipeline is coherent and well aligned with the observed Patch-Size Matching phenomenon.

Weaknesses:

1. Section 3 introduces the "Blindness in Patching" observation as a key motivation, but it would be helpful to include some experimental evidence or concrete examples to better support these initial claims.

2. While the method is framed as a black-box attack, it does rely on certain assumptions, such as the absence of cropping or content-based preprocessing.

3. The evaluation could benefit from broader validation. Currently, the Qwen-VL experiment is the main verifiable case for checking real patch size and resolution; extending analysis to more verifiable models would help confirm the generality of the findings.

---

> ### Author Rebuttal · Authors · 2026-03-31
>
> We thank the reviewer for the constructive comments. We address your concerns below.
>
> >  it would be helpful to include some experimental evidence or concrete examples to better support these initial claims.
>
> We appreciate the reviewer’s suggestion to strengthen the foundational motivation in Section 3. While the periodic accuracy collapses shown in Figure 1 and Section 5 serve as empirical evidence of the phenomenon, we agree that a concrete visualization of the "boundary erasure" mechanism would clarify the initial claim.
>
> Please refer to `Concrete Examples of the "Blindness in Patching"` in https://anonymous.4open.science/r/icml_figures-8962/README.md. We provide a simple task to prompt LLMs to find the size of a chessboard image, which is supposed to be very simple. However, flagship models such as GPT-5 and GPT-5.4 mini are not able to count the correct size of the chessboard even after reasoning.
>
> However, we noticed that GPT-5.4 is able to solve this question. We hypothesize that  GPT-5.4 may have undergone training on certain tasks that are sensitive to boundary information. such as computer-use.
>
> > It does rely on certain assumptions, such as the absence of cropping or content-based preprocessing.
>
> Thanks for raise this question. We acknowledge that our attacks require certain assumptions, however, we believe our assumptions  are realistic:
>
> **Cropping**: Flagship models often avoid aggressive cropping to maintain OCR and fine-grained detection performance. Furthermore, if cropping behavior is deterministic and can be "probed" using the shift-detection method (lines 92–93). Once the crop/offset is known, the adversary can pad the synthetic grid to re-align it with the patch grid, canceling the cropping,
>
> **Content-based preprocessing.**: Please refer to `GPT-4o results with detal=auto` in https://anonymous.4open.science/r/icml_figures-8962/README.md. Our results on detail=auto (which chooses between low/high based on input) confirm that the attack generalizes to models with content-based preprocessing
>
> **Black-Box Realism**: Compared to transfer-based attacks that assume surrogate-victim similarity, our "side-channel" approach relies only on the fundamental geometry of the ViT tokenizer, which is a nearly universal architectural choice in modern VLMs.
>
> > more verifiable evaluations on VLM to check the efficacy of the method?
>
> We thank the reviewer for the constructive suggestion to extend our validation to more verifiable models. We agree that broader evaluation strengthens the generality of our findings. We have updated our results to include more models and provide deeper evidence for the 'Blindness in Patching' observation.
>
> While many open-source models (e.g., Llama 3/4, Gemma 3) struggle with the grid-counting task across all cell sizes—likely due to lower inherent spatial reasoning capabilities, we have successfully verified our method on InternVL 3.5 30B.
>
> **Findings for InternVL 3.5**: Our Stage 2 (padding) analysis shows a periodic performance drop only when the image is padded to a canvas size of L=448, with an observed period of T=14. This indicates that InternVL 3.5 adopts a fixed-resolution preprocessing (target size 448) and a patch size of 14, consistent with documented configurations of similar high-performing vision encoders.
>
> **Stage 3 Consistency**: In our consistency-check attacks, the S=448 candidate showed a significantly higher winning rate compared to other candidates (e.g., 512), providing a second, independent layer of verification.
>
> The above results align with the open-source model's configuration.  Please refer to `InternVL results` in https://anonymous.4open.science/r/icml_figures-8962/README.md for the accuracy against cell sizes curves.

---

> > ### Author Rebuttal · Reviewer_6Eye · 2026-04-02
> >
> > Thanks for the response. I have no further questions.

---

### Official Review · Reviewer_vGPW · 2026-03-12

**Soundness:** 3
**Presentation:** 3
**Significance:** 2
**Originality:** 3
**Overall Recommendation:** 4
**Confidence:** 3

**Summary:**

The paper observes that ViT-style patchification in VLMs creates a task-level side channel that can be used to extract patch sizes. When synthetic colored grid images have cell sizes that are exact multiples of the hidden patch size, boundary cues vanish at tokenization, causing periodic accuracy collapses on a grid-counting task and allowing to extract this information from black-box deployments.

The authors propose a three-stage black-box attack to recover private preprocessing parameters from deployed VLMs: (1) sweep cell sizes on unpadded grids to detect dynamic-resolution models and estimate patch size P, (2) use zero-padded grids to restore periodicity under fixed-resolution models and recover the ratio S/P, and (3) apply a consistency check to identify the target resolution S. They validate on Qwen-VL, Claude, and GPT model families.

Finally, they show that the recovered parameters can improve model-targeted adversarial transfer attacks, where perturbations are crafted to fool one specific VLM while sparing another.

**Compliance With Llm Reviewing Policy:**

Affirmed.

**Final Justification:**

The authors resolved most of my concerns, I maintain my recommendation of weak accept.

**Key Questions For Authors:**

1. How do the GPT-4o results change under detail=high or detail=auto (the default)? Under high detail, the model uses multi-tile processing that differs fundamentally from the single fixed-resize model assumed by the paper. If the attack fails under default settings, this substantially limits practical relevance.

2. Have you compared against a simple baseline where the adversary enumerates all plausible (P, S) combinations from the small design space and selects whichever maximizes adversarial transfer? This comparison is critical for establishing that the side-channel extraction provides value beyond brute-force guessing.

**Limitations:**

Yes

**Strengths And Weaknesses:**

# Strengths

* The connection between patchification geometry and periodic task failures is well-formalized, and the derivation of when accuracy valleys occur follows cleanly from the patch grid structure. I found this to be the strongest aspect of the paper.
* The paper is clearly written and well-structured. I was able to follow the progression from phenomenon to methodology to experiments to downstream applicability without difficulty.
* The evaluation covers multiple model families (open-source Qwen for validation, proprietary Claude and GPT), which lends some generality to the PSM observation.

# Weaknesses

* I have concerns about the practical significance of the "model stealing" framing. The space of plausible configurations is very small: patch sizes are essentially {14, 16, 32}, target resolutions come from a handful of common values, and preprocessing (as considered in the paper) is either dynamic or fixed. An adversary could simply enumerate all plausible (P, S) pairs and pick the one that maximizes adversarial transfer, likely requiring fewer queries than the ~48,000 used here. The paper never compares against this "enumerate and test" baseline, which makes it hard to assess whether the side-channel approach provides any practical advantage.
* The GPT experiments are all conducted with detail=low, which is noted only in the appendix (lines 634-635). This is a significant issue for two reasons.
   * According to OpenAI's public API documentation, detail=low explicitly processes images at a fixed 512×512 resolution. This means the paper's multi-stage “discovery” that GPT-4o uses S=512 is recovering a publicly documented parameter, not a private architectural secret.
  * The default setting is auto, which typically selects high detail. Under high detail, GPT-4o uses an entirely different tile-based tokenization (as per the documentation, 512px tiles with aspect-ratio-preserving multi-scale processing), which fundamentally violates the paper's assumption of a single fixed-resolution resize. The attack's applicability under default settings is undemonstrated, and I suspect the multi-tile pipeline would break the approach.
* I have some concerns about the adversarial attack evaluation (Section 6).
  * The comparison with SSA-CWA and AnyAttack in Table 3 is not controlled. These methods use different surrogate ensembles, optimization procedures, and evaluation protocols. Including them in the same table creates a misleading impression of improvement. The only fair comparison is the paper's own Eq. 12 vs. Eq. 13.
  * There are no ablations on perturbation budget (\eps=8/255 is large), surrogate model choice (all three are CLIP-family ViT-H), or the interpolation method used to implement the preprocessing pipelines during optimization. Any of these could be driving the results more than the stolen parameters.
* The framework's generalizability to modern VLM architectures is limited. The attack assumes no cropping, square images, a static deterministic pipeline, and standard non-overlapping linear patch projection.

---

> ### Author Rebuttal · Authors · 2026-03-31
>
> We thank the reviewer for the constructive comments. We address your concerns below.
>
> ## Weakness
> 1. Practical significance and "enumerate and test" baseline.
>
> While the space of **standard configurations** is currently small, proprietary models are under no obligation to follow them. It is possible that certain proprietary model may use non-standard resolutions or patch sizes (via Neural Architecture Search) to optimize the performance-cost trade-off. Relying on "standard guesses" leaves an adversary blind to these optimizations.
>
> Regarding the "enumerate and test" baseline: we argue it is significantly less practical due to query and compute costs. To distinguish between similar configurations (e.g., P=32,S=512 vs. P=30,S=480, P=28,S=448) based on transferability, an adversary needs a large sample size for statistical significance. As shown in our new results below (following Table 3 settings), the gap in Attack Success Rate (ASR) is narrow (Claude 3.7 is no longer available at the time of rebuttal):
> | Attack method | GPT-4o | Claude 4.5 |
> |:-:|:-:|:-:|
> | Eq. (12) | 78.5 | 14.3 |
> | Eq. (13) Targeted to GPT-4o with P=32, S=512 | 86.5 | 5.1 |
> | Eq. (13) Targeted to GPT-4o with P=30, S=480 | 86.0 | 5.7 |
> | Eq. (13) Targeted to GPT-4o with P=28, S=448 | 86.3 | 6.5 |
>
> Distinguishing between 86.5% and 86.0% requires ~5,000 test images to reach statistical significance (p<0.05). If an adversary tests 10 plausible candidates, this requires 50k API calls and massive GPU overhead to generate 10 sets of adversarial examples. In contrast, our side-channel approach recovers parameters using a single set of synthetic grids. While our 48k calls provided a complete curve for the paper, discovery can be optimized to a few thousand calls by targeting specific cell sizes near expected multiples of P.
>
>
> 2. GPT experiments with detail=low vs. high/auto.
>
> The reviewer is correct that detail=low uses a documented 512×512 resolution. However, not all API providers may provide such information, and our methods do not need to take this information to draw the conclusion.
>
> Regarding detail=high/auto: We conducted additional experiments (see [Link] for plots). The attack remains successful. Even in multi-tile mode, GPT-4o processes a "global view" (a resized version of the original image) alongside high-res tiles. Our grid-based side channel triggers a collapse in this global representation, which is often sufficient to disrupt the model's high-level reasoning. The consistency of the accuracy drops across low, high, and auto modes demonstrates that the underlying patchification logic remains a viable side channel even in complex, multi-stage pipelines.
>
> 3. Concerns about the adversarial attack evaluation.
>
> **Controlled Comparison**: We acknowledge that SSA-CWA and AnyAttack use different ensembles. Our intent was not to claim a superior optimization algorithm, but to provide a proof-of-concept that knowing the victim's P and S allows a standard attack to significantly outperform a model-agnostic one. Comparing Eq. 12 vs. Eq. 13 serves as our controlled baseline.
>
> **Ablations**: We chose ϵ=8/255 and CLIP-family ViT-H surrogates because they are the standard benchmarks in recent literature (SSA, AnyAttack, M-Attack). While interpolation and budgets affect absolute ASR, they affect the surrogate and victim similarly; they do not explain the different performance changes GPT-4o and Claude 3.7  gained by matching the stolen parameters.
>
> 4. Generalizability to modern architectures.
>
> **Square images**: we only use square images for the ease of discussion, but **not** assume square images.
>
> **Cropping/Padding**: Flagship models often avoid aggressive cropping to maintain OCR and fine-grained detection performance. Furthermore, cropping behavior is deterministic and can be "probed" using the shift-detection method (lines 92–93). Once the crop/offset is known, the adversary can pad the synthetic grid to re-align it with the patch grid, canceling the cropping,
>
> **Dynamic Pipelines**: Our results on detail=auto (which chooses between low/high based on input) confirm that the attack generalizes to models with dynamic logic.
>
> ## Questions
>
> 1. **GPT-4o detail=high/auto:**
>
> Please refer to https://anonymous.4open.science/r/icml_figures-8962/ for the plot figures. The Acc for the same cell size may be different, however our conclusion remains the same and statically significant using the statical test discussed in the paper.
>
> 2. **Enumerate baseline:**
>
> As addressed in W1, the brute-force baseline is not more query-efficient. Our side-channel approach decouples "discovery" from "attack generation," making it a more efficient precursor to model-targeted adversarial attacks.

---

> > ### Author Rebuttal · Reviewer_vGPW · 2026-04-03
> >
> > Hi! Thank you for your detailed rebuttal, this has resolved most of my concerns.
> >
> > One small question, the link to detail experiments does not seem to be there. Is there someway to show this?

---

> > > ### Author Response · Authors · 2026-04-04
> > >
> > > Hi, are you able to open this link?
> > >
> > > https://anonymous.4open.science/r/icml_figures-8962/README.md
> > >
> > > We have already confirmed it with our colleagues that it can be opened. If it still won't open, can you try this link?
> > >
> > > https://github.com/icml26-3261/figures/blob/main/README.md

---

### Official Review · Reviewer_oXvz · 2026-03-13

**Soundness:** 3
**Presentation:** 2
**Significance:** 3
**Originality:** 3
**Overall Recommendation:** 4
**Confidence:** 4

**Summary:**

The paper proposes a black-box model-stealing attack against VLMs, aimed at recovering private architectural and preprocessing parameters. Specifically, the visual patch size and internal image resizing strategy (dynamic vs. fixed resolution). The paper reveals the "Patch-Size Matching" (PSM) phenomenon: when a multi-colored grid image is input to a VLM, if the grid cell size perfectly aligns with the model's hidden patch size, edge cues are erased during tokenization, causing a periodic drop in grid-counting accuracy. Based on this, it designs a systematic three-stage probing method (unpadded sweep, zero-padding to restore periodicity, and a relative answer consistency check), successfully deriving these key parameters for open-source models like Qwen-VL and closed-source models like GPT-4o, GPT-4.1-mini, and Claude 3.7/4.5.

**Compliance With Llm Reviewing Policy:**

Affirmed.

**Final Justification:**

The authors' responses have well addressed my concerns. I raise my score accordingly.

**Key Questions For Authors:**

•	Assumption robustness: How does the attack perform against models using multi-scale routing or aggressive random cropping, and under what conditions would it fail?

•	Ablation of contrastive objective: Can you provide results for an attack optimized solely against the target pipeline, without the contrastive term, to isolate its contribution?

•	Task generalizability: How transferable is the probing method to VLMs that may not reliably perform the grid-counting task, or to other general vision tasks?

•	Query cost analysis: Can you provide a detailed analysis of total query cost and discuss strategies to optimize it for rate-limited APIs?

•	Updated baselines: Could you compare your attack against more recent works like M-Attack and FOA-Attack?

•	Minor correction: Can you resolve the reference error in Section 5.1 (line 298) to ensure clarity?

**Limitations:**

yes

**Strengths And Weaknesses:**

**Strengths:**

1.	The proposed three-stage probing method is correct, although of incremental novelty.

2.	The authors validate their attack on a diverse set of models, which demonstrates effectiveness of the proposed method.

**Weaknesses:**

1. Assumption limitations: The probing attack relies on specific preprocessing assumptions (e.g., symmetric resizing, no content-adaptive cropping). As the authors acknowledge, if a system uses multi-scale routing or aggressive random cropping, the method may fail.

2. Unclear contribution of the contrastive learning component: While the feasibility of targeted attacks is meaningful, the contribution of the contrastive learning component is unclear. The paper lacks experimental results for optimizing solely against a single target model without utilizing the contrastive objective.

3. Narrow Focus on Counting Task: The entire attack is built upon the model's ability to perform a specific "grid-size counting" task. You should conduct evaluations on more other tasks.

4. Scalability and Cost: The attack requires a significant number of queries (e.g., M=200 random grid instances per D, sweeping D over a range, multiple canvas sizes L). While this is feasible for a determined adversary, the paper could provide a more detailed analysis of the query cost and how it might be optimized. For a very large model or one with rate-limiting, this could be a practical barrier.

5. Insufficient evaluations. For the results on Table 3, authors only compare the methods with SSA-CWA and AnyAttack, which are somewhat out of the date. You should compare with more recent works like M-attack [1] and FOA-attack [2].
[1] Zhaoyi Li, Xiaohan Zhao, Dong-Dong Wu, Jiacheng Cui, and Zhiqiang Shen. A frustratingly simple yet highly effective attack baseline: Over 90% success rate against the strong black-box models of gpt-4.5/4o/o1. NeurIPS 2025.
[2] Xiaojun Jia, Sensen Gao, Simeng Qin, Tianyu Pang, Chao Du, Yihao Huang, Xinfeng Li, Yiming Li, Bo Li, Yang Liu. Adversarial Attacks against Closed-Source MLLMs via Feature Optimal Alignment. NeurIPS 2025.

6. Minor typos: In Section 5.1, there is an unresolved reference error on line 298.

---

> ### Author Rebuttal · Authors · 2026-03-30
>
> We thank the reviewer for the constructive comments. We address your concerns below.
>
> 1.
>
> `Assumption limitations.`
> We do **not** assume symmetric resizing for arbitrary inputs; we only assume that a square input remains square after preprocessing. For content-adaptive cropping, we provide additional GPT-4o results under *detail=high* and *detail=auto*. Under these model, GPT-4o would apply content-adaptive cropping, and our results are still valid. Please refer to https://anonymous.4open.science/r/icml_figures-8962/ for the plot figures. Our “no cropping” assumption is realistic for flagship models that must preserve information for OCR and object detection.
>
> `How does the attack perform against models using multi-scale routing or aggressive random cropping?`  Our new GPT-4o (*detail="auto"*) results show robustness to multi-scale routing. If the model uses **random** cropping, our method returns a non-identifiable result rather than a wrong prediction. If it uses **deterministic** cropping, we can probe the cropping behavior using the method in lines 92–93 and then pad inputs to cancel the shift. Our new results show robustness to deterministic cropping; we will add them in the revision.
>
> `Under what conditions would it fail?` The method may fail when the model uses randomized preprocessing, when the model cannot reliably solve the grid-counting probe itself, or when the model employs a non-ViT vision encoder.
>
> 2.
>
> `Unclear contribution of the contrastive learning component.` The contribution of the contrastive objective is **model selectivity**, not merely stronger attack strength. Eq. (12) is a model-agnostic transfer baseline, while Eq. (13) is a pipeline-aware targeted objective that increases ASR on the target model and suppresses transfer to the non-target model. Thus, the gap between Eq. (12) and Eq. (13) already supports our claim that stolen preprocessing information enables model-targeted manipulation, including benchmark manipulation.
>
> `The paper lacks results for optimizing solely against a single target model without the contrastive objective.` We add this ablation below:
>
> | Attack method | GPT-4o | Claude 3.7 |
> |:-|:-:|:-:|
> | Eq. (12) | 78.5 | 13.5 |
> | Eq. (13), both terms, targeted to GPT-4o | 86.5 | 4.5 |
> | Eq. (13), first term only, targeted to GPT-4o | 87.3 | 11.8 |
> | Eq. (13), both terms, targeted to Claude 3.7 | 62.5 | 18.5 |
> | Eq. (13), first term only, targeted to Claude 3.7 | 72.5 | 19.3 |
>
> These results show that the contrastive term mainly improves **selectivity** by suppressing transfer to the non-target model.
>
> 3.
>
>  `Narrow focus on the counting task.` We respectfully disagree with this framing. Our claim is not that all VLM tasks exhibit the same phenomenon. Rather, grid-size counting is a boundary-sensitive **probe** that cleanly exposes the patch-alignment side channel. It is the probing instrument, not the end goal. We also show robustness across different grid sizes \(N\) in Appendix B, and the downstream consequence is evaluated separately via model-targeted adversarial attacks in Sec. 6.
>
> 4.
>
>  `Scalability and cost.` As discussed in Sec. 5.3, about 200 images per cell size are sufficient to make the phenomenon visible, i.e., about 8K images for one configuration. If we test 5 candidate target sizes, this totals about 40K API calls. Since each query asks the model to answer directly, the output is typically only 200–300 tokens; even using a conservative upper bound of 400 tokens, this is about 8M tokens in the worst case for one target model.
>
> `discuss strategies to optimize` The above estimation is the worst case. In practice, we can test fewer number of images for most candidate cell sizes. As we show the confidence intervals in the figures, we only need to test more images for one cell size if its confidence interval does not overlap with its neighbor cell sizes' confidence interval.
>
> 5.
>
>  `Insufficient evaluations.` Our goal in Table 3 is not to claim the strongest generic black-box attack, but to show that leaked preprocessing information enables **model-selective** manipulation. Therefore, the key comparison is between the model-agnostic baseline in Eq. (12) and the pipeline-aware targeted objective in Eq. (13).
>
> We also add M-attack for reference. Note that M-attack uses a different setting from the original NIPS2017 benchmark: it randomly selects another MSCOCO image as target. We keep the original benchmark setting and use the stricter evaluation protocol in Appendix D1.
>
> | Attack method | GPT-4o | Claude 3.7 |
> |:-|:-:|:-:|
> | Eq. (12) | 78.5 | 13.5 |
> | M-attack | 32.8 | 4.8 |
>
> 6. `Unresolved reference on line 298.` Thank you; this should be Figure 8.

---

> > ### Author Rebuttal · Reviewer_oXvz · 2026-04-03
> >
> > The authors' responses have well addressed my concerns.
> > Regarding Point 3, I would like to clarify my original intent: I was questioning whether this probing method remains effective for models that cannot reliably solve the grid-counting task. I note that the authors have actually answered this exact question at the end of their response to Point 1, where they acknowledged that the method may fail in such scenarios. I suggest explicitly adding this limitation to the final manuscript.

---

### Official Review · Reviewer_tjwh · 2026-03-14

**Soundness:** 4
**Presentation:** 3
**Significance:** 4
**Originality:** 4
**Overall Recommendation:** 5
**Confidence:** 4

**Summary:**

The paper focuses on stealing private vision tokenizer configurations from deployed VLMs.
It infers patch size by sweeping grid sizes and detecting periodic accuracy drops.
Padding and consistency checks further reveal preprocessing details and target resolution.
The experimental results confirm reliable parameter extraction across various models, enabling effective downstream attacks.

**Compliance With Llm Reviewing Policy:**

Affirmed.

**Final Justification:**

The paper studies a challenging task, proposes an elegant probing method, and demonstrates strong empirical performance. I have no other concerns. Therefore, I maintain my rating of Accept.

**Key Questions For Authors:**

See in the Weakness.

**Limitations:**

Yes

**Strengths And Weaknesses:**

**Pros**
1. The paper investigates a meaningful and challenging problem: detecting the patch size and preprocessing strategies of visual encoders in vision-language models (VLMs).
2. It cleverly leverages the "visual strawberry" phenomenon present in current VLMs, using a counting grids task to probe the visual encoder's patch size and preprocessing methods. The approach is both simple and ingenious.
3. The experimental results on both open-source and closed-source VLMs are consistent with the proposed hypotheses and methods, allowing for high-confidence detection of visual encoder information.


**Cons**
1. Regarding models like Qwen3-VL and Claude, which support native resolution inputs, why does the counting grids task still achieve a certain probability of correct predictions when the cell size is a multiple of the patch size? Moreover, in some models, accuracy tends to increase as the cell size grows, as shown in Figure 1.
2. There is a typo in Line 289: “Appendix ?” should be corrected.

---

> ### Author Rebuttal · Authors · 2026-03-31
>
> We thank the reviewer for the constructive comments. We address your concerns below.
>
> 1. why does the counting grids task still achieve a certain probability of correct predictions when the cell size is a multiple of the patch size?
>
> **Color Semantic Cues**: The primary reason accuracy remains above zero is that the model can leverage the color distribution of the patches. Even if cell boundaries are "erased" at the token level, the model still receives tokens of different colors. If a model sees a sequence of blue, red, and green tokens, it can still infer the presence of multiple cells through color-based reasoning.
>
> **Difficulty Calibration**: As noted in the paper, we intentionally used a palette of $\mathcal{C}=8$ colors. We found that using a large number of distinct colors creates a "shortcut" that makes the task trivial regardless of patch alignment. We calibrated the palette size specifically so that the model achieves non-trivial baseline accuracy but still exhibits sharp collapses when the most reliable cue, i.e., the boundaries, is removed
>
> 2. In some models, accuracy tends to increase as the cell size grows, as shown in Figure 1.
>
> This is expected. The upward trend in accuracy as D increases is expected and stems from the Interior-to-Boundary Token Ratio. As D grows, the number of "clean" interior patches (those not straddling a boundary) increases quadratically . These unambiguous tokens provide a stronger signal for counting.
>
> 3. There is a typo in Line 289: “Appendix ?” should be corrected.
>
> Thank you. We will fix this in a new version. The missing reference should be Figure 8.

---

> > ### Author Rebuttal · Reviewer_tjwh · 2026-04-03
> >
> > Thank you for your response. I have no further questions. I will keep the rating.

---

### Decision · Program_Chairs · 2026-04-30

**Decision:**

Accept (regular)

**Comment:**

The paper identifies a novel and security-relevant side-channel in VLMs, leveraging patchification artifacts to recover private preprocessing parameters via a well-designed black-box attack. Reviewers agree that the proposed three-stage probing methodology is intuitive, technically sound, and supported by consistent empirical results across both open- and closed-source models. The connection between patch geometry and periodic task failures is particularly insightful and represents a meaningful contribution to understanding VLM vulnerabilities. While concerns remain regarding assumptions on preprocessing pipelines, query efficiency, and limited comparisons or broader evaluations, these do not undermine the core contribution. Several reviewers noted that the authors adequately addressed key concerns during rebuttal, improving clarity and strengthening empirical validation. Overall, given the novelty, practical relevance, and solid experimental support, the paper is accepted.